

# Phenotypic and genetic diversity of doubled haploid bread wheat population and molecular validation for spike characteristics, end-use quality, and biofortification capacity

Imren Kutlu[1], Sadettin Çelik[2], Yaşar Karaduman[3] and Özcan Yorgancılar[4]

[1] Department of Field Crops, Faculty of Agriculture, Osmangazi University, Eskişehir, Turkey
[2] Department of Forestry, Genç Vocational School, Bingöl University, Bingöl, Turkey
[3] Department of Food Engineering, Faculty of Agriculture, Osmangazi University, Eskişehir, Turkey
[4] Department of Biotechnology, Transitional Zone Agricultural Research Institute, Eskişehir, Turkey

Corresponding author
Imren Kutlu, ikutlu@ogu.edu.tr

## ABSTRACT

Increasing grain quality and nutritional value along with yield in bread wheat is one of the leading breeding goals. Selection of genotypes with desired traits using traditional breeding selection methods is very time-consuming and often not possible due to the interaction of environmental factors. By identifying DNA markers that can be used to identify genotypes with desired alleles, high-quality and bio-fortified bread wheat production can be achieved in a short time and cost-effectively. In the present study, 134 doubled haploid (DH) wheat lines and their four parents were phenotypically evaluated for yield components (spike characteristics), quality parameters, and grain Fe and Zn concentrations in two successive growing seasons. At the same time, ten genic simple sequence repeats (SSR) markers linked to genes related to the traits examined were validated and subsequently used for molecular characterization of trait-specific candidate genotypes. Significant genotypic variations were determined for all studied traits and many genotypes with desired phenotypic values were detected. The evaluation performed with 10 SSR markers revealed significant polymorphism between genotypes. The polymorphic information content (PIC) values of 10 markers ranged from 0.00 to 0.87. Six out of 10 SSRs could be more effective in representing the genotypic differentiation of the DH population as they demonstrated the highest genetic diversity. Both Unweighted Pair Group Method with Arithmetic Mean (UPGMA) clustering and STRUCTURE analyses divided 138 wheat genotypes into five (K = 5) main groups. These analyzes were indicative of genetic variation due to hybridization and segregation in the DH population and the differentiation of the genotypes from their parents. Single marker regression analysis showed that both Xbarc61 and Xbarc146 had significant relationships with grain Fe and Zn concentrations, while Xbarc61 related to spike characteristics and Xbarc146 related to quality traits, separately. Other than these, Xgwm282 was associated with spike harvest index, SDS sedimentation value and Fe grain concentration, while Gwm445 was associated with spikelet number, grain

number per spike and grain Fe concentration. These markers were validated for the studied DH population during the present study and they could be effectively used for marker-assisted selection to improve grain yield, quality, and bio-fortification capacity of bread wheat.

## INTRODUCTION

Wheat, the oldest cultivated and the most widely grown crop, is defined as the king of cereals because it has important place in international grain trade. Wheat is a significant micronutrient source, providing 20% of the calories and 25% of the proteins daily consumed in human nutrition. Therefore, it has an important role ensuring the zero hunger committed within the scope of the Sustainable Development Goals and guaranteeing food and nutrition security (*Shewry & Hey, 2015*; *Sendhil et al., 2022*). Improving the nutritional quality of the grain such as protein and micronutrient levels, along with the grain yield and alleviating malnutrition and hidden hunger are the main goals of breeding programs (*Lantican et al., 2016*). Major improvements in agriculture and plant breeding in the 20[th] century have brought important improvements in the end-use quality, processing, and yield of the most widely grown agricultural products (*Bradshaw, 2016*). Despite these advances, food scarcity is still widespread. It is estimated that there are around 800 million malnourished people in the world, most of them in developing countries. Besides, hidden hunger and malnutrition are quite common, and more than half of the population of the world suffers from micronutrient deficiency, especially iron (Fe) and zinc (Zn) (*Zhao & Mcgrath, 2009*; *FAO, 2015*).

Genetic bio-fortification is a strategy that targets higher micronutrient levels in cereal grains, higher bioavailability of micronutrients, and higher intestinal absorption. Plant breeding aims to produce new cultivars with higher micronutrient levels for this purpose (*Bouis et al., 2011*). Current breeding approaches intend to supply a sustainable and cost-effective remedy to malnutrition problems by surveying natural genetic diversity to develop cultivars with high mineral content (*Sharma, Aggarwal & Kaur, 2017*).

Genetic diversity in populations obtained through hybridization is considered to originate from the parents. However, the flexibility of the genome and the selection response capacity to, transgressive segregation, base changes, epigenetic changes, or inhibitive gene effects also greatly affect the phenotype and they can create new alleles through their mutual interactions. Thus, the genetic variation may not only be caused by the parents (*Rasmusson & Phillips, 1997*). Newly derived alleles are an important source of variation. The expansion of the phenotypic boundaries of the traits together with the increasing inhibiting gene effect may also be an unpredictable and independent variation within the doubled haploids (DHs) of genotypes obtained from long-term selection, with continuous genetic gain from the narrowing genetic pools. In this case, genetic progress may be coming from the original variation as well as to some extent from the increased

covered gene effect among gene combinations of novel origin (*Rasmusson & Phillips, 1997*). The ability of parental genotypes to maintain their genetic potential in their offspring may have a significant impact on the expansion and diversification of observed changes. Increasing the genetic diversity desired for quantitative traits will accelerate progress in genetic studies for understanding the genetic structure of complicated traits.

Using molecular markers for the evaluation of genetic diversity instead of traditional phenotypic traits makes it faster and cheaper to study. In addition, they could show a high-level of polymorphism that could be detectable in all tissues. Generally, SSR markers are presumed appropriate for studies of genetic mapping, marker-assisted selection, and genetic diversity in wheat (*Boopathi, 2020*). Genetic diversity and mapping of yield and quality traits of wheat species and wild relatives have been studied quite commonly and SSR markers linked to these traits detected for use marker-assisted selection of genotypes with high yield and quality (*Monasterio & Graham, 2000*; *Cakmak et al., 2004*; *Chatzav et al., 2010*; *Blanco et al., 2012*; *Phougat & Sethi, 2019*; *Ahmed et al., 2020*; *An et al., 2022*; *Kumar, Gill & Nagarajan, 2022*; *Shamsabadi et al., 2022*). However, it has been determined that the closely related markers identified in these studies have phenotypic characteristics that vary according to populations. This has made it necessary to characterize wheat genotypes for yield, quality and bio-fortification and to validate predefined markers associated with the aforementioned traits attributed in many literatures. Therefore, this study was conducted to evaluate a DH wheat population for traits related to high yield, quality, and bio-fortification followed by molecular characterization using trait-specific/genic SSR markers. Efforts were also made to validate some predefined marker-trait relationships for high yield, quality and bio-fortification in wheat.

# MATERIALS AND METHODS

## Plant material and growth conditions

In the experiment, 134 DH bread wheat lines developed by anther culture technique from the $F_2$ generation of the "Vratza/Kate(8)*2//WuGeng8025/3/Sonmez" hybrid, and four parents were used as the material (Fig. 1).

Doubled haploid lines were obtained from a joint study in the Biotechnology Laboratory of the Transitional Zone Agricultural Research Institute, and seeds were reproduced under controlled conditions. Doubled haploid lines and their parents (as control cultivars) were tested in an augmented trial design for two seasons (2017–2018 and 2018–2019) in Eskisehir conditions. The research area is between 39°45′36.45″–39°45′36.09″ N and 30°28′10.59″–30°28′11.25″ E geographical coordinates. The soil at the trial location is clayey, slightly alkaline (pH 7.9–8.3), salt-free, low organic matter (1.2%), and moderately limy (9.6%). The ratios of phosphorus ($P_2O_5$), potassium ($K_2O$), Fe, and Zn available to the plant are 49.2, 2,423.1 kg/ha, 3.94, and 0.21 ppm, respectively. The total precipitation during the first experiment year was 479 mm, the average temperature was 11.86 °C and the average humidity was 82.12%, and these values realized as 406 mm, 10.73 °C, and 75.04% the second year, respectively.

The plots consist of two rows, each two meters long with 30 cm between rows and 5 cm in row spaces, and a distance of 2 m has been left between the plots. A fixed fertilizer dose

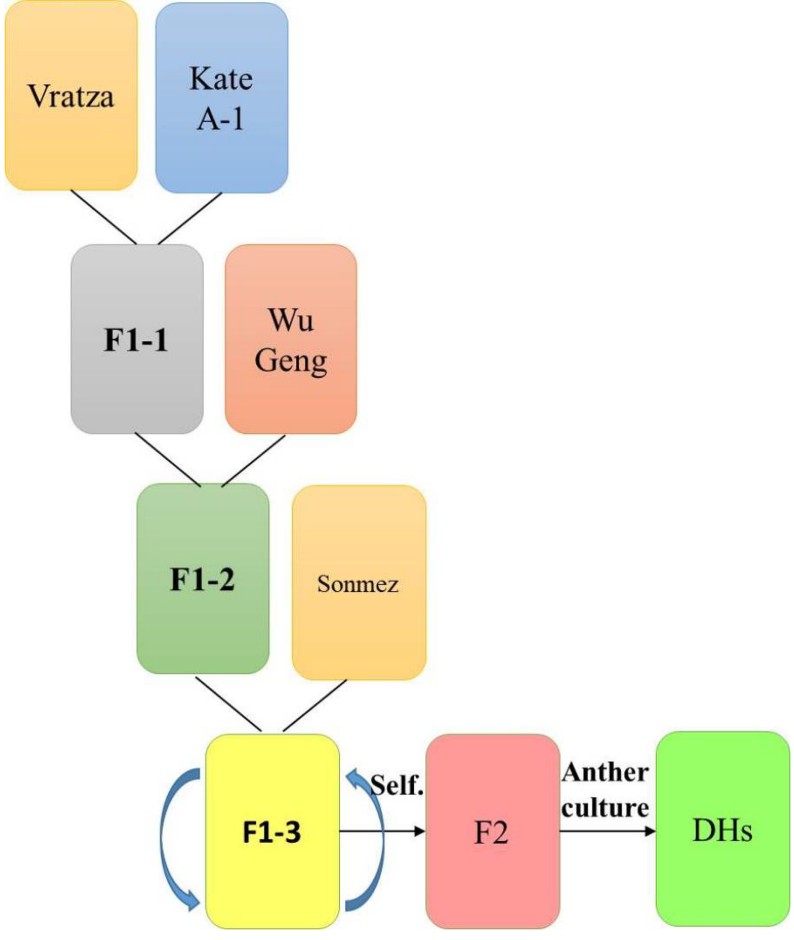

**Figure 1 Schematic representation of how the four-parental DH population were obtained.**

of 120 kg/ha N and 100 kg/ha $P_2O_5$ was applied to the plots, and all of the $P_2O_5$ was given in sowing, half of the N in sowing, and the other half of N in the tillering stage (Zadoks scale 23). Supplementary irrigation was carried out twice, during the stem elongation and flowering stages, as 300 mm in total.

## Phenotypic measurements

The plants were harvested at maturity by handpicking the spikes from the plots. Spike characteristics such as spike length, spike weight, number of spikelets, number of grains per spike, grain weight per spike, spike density, and spike harvest index were measured. Thousand-grain weight and grain hardness were measured with SKCS 4100 (Perten Instruments, North America, Inc., Springfield, IL, USA) on 300 grains of each sample according to AACC Method 55-31 (*American Association of Cereal Chemists (AACC), 2002*). The protein rate and gluten content (%) were determined using a NIR spectroscopy device (NIR 6500; Foss, Hillerod, Denmark) according to AACC method 39-01.01 (*American Association of Cereal Chemists (AACC), 2010*), while the SDS sedimentation value was found following the procedure of *Peña et al. (1990)*.

Zinc and Fe concentrations were determined by atomic absorption spectrophotometer (NovAA 350; Analytik jena, Germany), after digesting 0.3 g samples in a closed microwave system (CEM Mars6) with $HNO_3$ (*Soil Survey Laboratory Methods, 2004*).

## Statistical analysis of phenotypic data

All phenotypic variables were tested with analysis of variance (ANOVA) using unbalanced designs (augmented) and calculated descriptive statistics for the data of each trait and year were gained. The average of the two-year values of each trait was used for the normal distribution and histogram. A principal component analysis (PCA) based on a correlation matrix was applied to determine the associations among the 14 studied traits. The first five factors, which had eigenvalues greater than 1.0, were considered effective for interpreting the results. The maximum number of iterations for convergence was 25, and the variamax rotation method was applied. The first two components, which have most of the variability, were presented as bi-plot ordinations of DH populations and traits studied. These analyses were carried out with the GenStat v.12 software (*VSN, 2022*).

## DNA isolation, PCR amplification, and visualization of markers

Genomic DNA was extracted from 100 mg of fresh frozen leaves of individual plants for all genotypes grown in the plastic pot in a greenhouse using Gene Jet Plant Genomic DNA Purification Mini Kits (K0791; Thermo Fisher Scientific, Waltham, MA, USA).
The quantity and quality of the extracted DNA were determined at 260 and 280 nm using a Thermo Scientific NanoDrop 2000™ spectrophotometer. Ten SSR markers, presented in Table 1, were selected and synthesized according to the information available in the Grain Genes database (https://graingenes.org/GG3/). These markers were randomly distributed across the wheat genome and mapped QTLs linked to specific markers from previous studies were selected (*Song et al., 2005*; *Marza et al., 2006*; *Genc et al., 2009*; *Peleg et al., 2009*). Polymerase chain reactions (PCR) were performed in a Thermal Cycler (Thermo Fisher Scientific, Waltham, MA, USA) in a volume of 30 μL of the master mix containing: 1 μL of DNA template (50 ng/mL), 23.21 μL of $ddH_2O$, 3 μL of 10x PCR buffer with $MgCl_2$, 0.24 μL dNTPs, 0.15 μL of each forward and reverse primers, and 0.45 μL of Taq polymerase. The amplification steps were as follows: 1 cycle at 95 °C for 5 min, then 35 cycles comprising 95 °C for 0.5 min, annealing of the primer at 55 °C for 1 min, and then extension at 72 °C for 1 min. The final extension was carried out at 72 °C for 10 min. The amplification products were electrophoresed on 2% agarose gels and for staining, 3 μL RedSafe was added to each sample. Thermo Ladder (SM1103) was used to compare the sizes of SSR products. Gel scanning was performed using the Kodak Gel Logic 200-Imaging System (Kodak, NY, US). Phoretix 1D Pro gel analysis software (TotalLab Ltd, Newcastle upon Tyne, UK) was used to estimate fragment size as a base pair.

## Analysis of genetic diversity and population structure

The analysis of genetic diversity was performed based on each of the SSR primer profiles by using binary data of the amplified SSR products. The genetic parameters such as the number of alleles (Na), number of effective alleles (Ne), Shannon's information index (I),

**Table 1 The SSR markers used in the study.**

| SSR markers | Sequence of primers | Chr | Allele size (bp) |
|---|---|---|---|
| Gwm445 | 5'TTTGTTGGGGGTTAGGATTAG3'<br>5'CCTTAACACTTGCTGGTAGTGA3' | 2A | 227–256 |
| Xbarc108 | 5'GCGGGTCGTTTCCTGGAAATTCATCTAA3'<br>5'GCGAAATGATTGGCGTTACACCTGTTG3' | 7A | 201–251 |
| Xbarc146 | 5'AAGGCGATGCTGCAGCTAAT3'<br>5'GGCAATATGGAAACTGGAGAGAAAT3' | 6A | 90–163 |
| Xbarc149 | 5'ATTCACTTGCCCCTTTTAAACTCT3'<br>5'GAGCCGTAGGAAGGACATCTAGTG3' | 1D | 202–336 |
| Xbarc61 | 5'TGCATACATTGATTCATAACTCTCT3'<br>5'TCTTCGAGCGTTATGATTGAT3' | 1B | 57–164 |
| Xgwm154 | 5'TCACAGAGAGAGAGGGAGGG3'<br>5' ATGTGTACATGTTGCCTGCA3' | 5A | 198 |
| Xgwm282 | 5'TTGGCCGTGTAAGGCAG3'<br>5'TCTCATTCACACACAACACTAGC3' | 7A | 216–364 |
| Xgwm473 | 5'TCATACGGGTATGGTTGGAC3'<br>5'CACCCCCTTGTTGGTCAC3' | 7A-L | 206 |
| Xgwm515 | 5'AACACAATGGCAAATGCAGA3'<br>5'CCTTCCTAGTAAGTGTGCCTCA3' | 2A | 70–94 |
| Xwmc177 | 5'AGGGCTCTCTTTAATTCTTGCT3'<br>5'GGTCTATCGTAATCCACCTGTA3' | 2A | 184 |

observed heterozygosity (Ho), and expected heterozygosity (He) were calculated using the GenAlEx software version 6.5 (*Peakall & Smouse, 2012*). Polymorphic information content (PIC), identified as the most suitable marker for genetic diversity evaluation, was determined according to *Anderson et al. (1993)* for each SSR marker. The dendrogram based on the relationship matrix was generated with the Unweighted Pair Group Method with Arithmetic Mean (UPGMA) method using the MEGA7 software (*Kumar, Stecher & Tamura, 2016*).

The genetic structure of the population was analyzed by taking probable sub-populations (K) and higher delta K-value using STRUCTURE v2.3.6 software (*Perrier & Jacquemoud-Collet, 2006*). To estimate population structure a model with 100,000 iterations, a 100,000 burn-in period, and a K value run of 10 times were used. The true number of clusters was estimated according to the procedure described by *Evanno, Regnaut & Goudet (2005)* using STRUCTURE HARVESTER (http://taylor0.biology.ucla.edu/structureHarvester).

## Analysis of marker-trait association through simple regression analysis

Simple regression analysis was done with each phenotypic and polymorphic SSR marker's data using the GenStat v.12 package program (*VSN, 2022*). Genotypic data from SSR markers were used as an independent variable (X-variable) and phenotypic data as a dependent variable (Y-variable). When the significance of the regression coefficient was

0.05, it was accepted that there was a potential relationship between the marker and the traits.

## RESULTS

### Phenotypic assessment

The genotypes showed significant phenotypic variation for all the traits examined. The years also indicated significant results for most of the traits except grain number per spike and spike harvest index. The genotype-by-year interaction showed significant results for all of the traits. The mean values for the spike characteristics were generally close to each other in both years. The second year values were high for the spike length and spike density, however low for others. On the contrary, the quality characteristics were higher in the second year except for the Zn concentration. When the CV, Skewness and Kurtosis values are examined, it is seen that there is a moderate and high variability and the values show a normal distribution (Table 2). Histogram graphs showing the distribution of 134 DH lines and parents in terms of spike characteristics were given in Fig. 2. The genotypes showed a distribution close to normal and there was variation between the lines. The spike length of genotypes had a mean of 8.58 cm (minimum 3.95 cm and maximum 11.60 cm), while spike weight ranged from 0.82 to 3.03 g. Moreover, the number of spikelets and grain per spike had an average of 16.46 and 37.24, respectively. Furthermore, the mean grain weight per spike ranged from 0.61 to 2.05 g with an average of 1.43 g. The average spike harvest index and spike density were 66.71% and 52.49, respectively. The mean thousand-grain weight was 36.62 g with a range from 29.86 to 46.06 g.

The phenotypic data for SKCS hardness index, protein rate, SDS sedimentation value, gluten content, Fe, and Zn concentrations followed the normal distribution, suggesting that these traits were controlled by multiple loci (Fig. 3). The SKCS hardness index had an average of 57.79 (minimum 21.91 and maximum 96.22) with a 10.66 standard deviation, and the protein rate ranged from 12.30% to 17.82% with a 0.83% standard deviation. The grain Fe and Zn concentrations of the wheat genotypes used in this study vary between 18.37–52.51 and 16.66–51.24 mg/kg, respectively.

Principal component analysis (PCA) extracted five principal components that had Eigenvalues greater than one and explained 75.56% of the variation (Table 3). The PC1 that explained 25.56% of the variation among genotypes was loaded positively with all spike characteristics, and the spike weight, number of spikelets, grain number, and weight per spike had the highest factor loadings. The protein rate and gluten content that explained 15.56% of the genotypic variation were PC2 (Table 4). All examined parameters except spike harvest index, SDS sedimentation value, and grain Fe content contributed maximum towards the total variability present in the evaluated genotypes. Genotypes are difficult to distinguish as they are clustered in the center of the graph (Fig. 4). However, genotypes numbered 19, 29, 48, and 83 were reflected much better performance compared to the rest of the genotypes for thousand-grain weight, protein rate, gluten content, grain Fe, and Zn concentrations. The genotypes numbered 8, 31, 96, and 105, had high spike measures, and can be specifically selected to breed for high-yield capacity.

**Table 2 ANOVA and descriptive statistics for examined characters.**

| Characters | Year | Minimum | Maximum | Means ± SE | CV% | Skewness | Kurtosis | Genotype (G) | Year (Y) | Y × G |
|---|---|---|---|---|---|---|---|---|---|---|
| | | | | Descriptive statistics | | | | LSDs ($p > 0.05$) | | |
| Spike length (cm) | 1 | 4.30 | 13.30 | 8.02 ± 0.11 | 15.52 | 0.92 | 4.12 | 0.31** | 0.03** | 0.42** |
| | 2 | 3.60 | 12.67 | 9.16 ± 0.10 | 12.93 | −0.52 | 2.92 | | | |
| Spike weight (g) | 1 | 0.80 | 3.01 | 2.15 ± 0.03 | 16.84 | −0.18 | 0.53 | 0.11** | 0.01* | 0.15** |
| | 2 | 0.84 | 3.24 | 2.13 ± 0.03 | 18.76 | −0.03 | 0.75 | | | |
| Number of spikelet | 1 | 10.77 | 19.93 | 17.24 ± 0.13 | 8.76 | −0.80 | 1.41 | 0.55** | 0.06** | 0.75** |
| | 2 | 12.33 | 18.92 | 15.67 ± 0.11 | 8.20 | −0.14 | −0.19 | | | |
| Grain number per spike | 1 | 19.60 | 52.33 | 37.54 ± 0.62 | 19.40 | −0.18 | −0.76 | 3.85** | 0.41ns | 5.23** |
| | 2 | 21.80 | 60.80 | 36.94 ± 0.58 | 18.44 | 0.32 | 0.47 | | | |
| Grain weight per spike (g) | 1 | 0.56 | 2.03 | 1.44 ± 0.02 | 19.49 | −0.09 | −0.26 | 0.14** | 0.02** | 0.20** |
| | 2 | 0.67 | 2.21 | 1.42 ± 0.02 | 20.01 | 0.18 | 0.54 | | | |
| Spike harvest index (%) | 1 | 55.51 | 78.15 | 67.04 ± 0.39 | 6.87 | −0.21 | −0.36 | 6.83** | 0.73ns | 9.28** |
| | 2 | 38.30 | 79.34 | 66.37 ± 0.47 | 8.36 | −1.30 | 4.25 | | | |
| Spike density | 1 | 35.84 | 85.98 | 46.61 ± 0.58 | 14.68 | 2.07 | 8.81 | 0.55** | 0.06** | 0.75** |
| | 2 | 27.41 | 74.31 | 58.37 ± 0.58 | 11.74 | −0.91 | 3.40 | | | |
| Thousand grain weight (g) | 1 | 25.60 | 55.56 | 37.74 ± 0.40 | 12.56 | 0.14 | 0.65 | 4.33** | 0.46** | 5.88** |
| | 2 | 25.72 | 47.24 | 35.51 ± 0.32 | 10.47 | 0.31 | 1.30 | | | |
| SKCS value (HI) | 1 | 22.21 | 128.79 | 46.47 ± 1.19 | 31.64 | 1.36 | 5.90 | 3.66** | 0.39** | 4.97** |
| | 2 | 21.20 | 95.53 | 69.10 ± 1.08 | 18.40 | −1.64 | 3.22 | | | |
| Protein rate (%) | 1 | 12.01 | 18.22 | 14.52 ± 0.08 | 6.61 | 0.47 | 1.27 | 0.58** | 0.06** | 0.79** |
| | 2 | 11.77 | 18.87 | 15.29 ± 0.09 | 6.95 | 0.21 | 0.60 | | | |
| Sedimentation value (ml) | 1 | 5.00 | 17.00 | 9.96 ± 0.18 | 21.49 | 0.22 | 0.05 | 2.20** | 0.24** | 2.99** |
| | 2 | 6.92 | 20.50 | 13.61 ± 0.29 | 28.06 | −0.02 | −1.00 | | | |
| Gluten content (%) | 1 | 22.52 | 44.20 | 32.69 ± 0.27 | 9.83 | 0.23 | 1.23 | 2.36** | 0.25** | 3.21** |
| | 2 | 18.32 | 42.29 | 34.54 ± 0.29 | 10.05 | −0.47 | 2.40 | | | |
| Grain Fe conc. (mg kg$^{-1}$) | 1 | 11.51 | 74.95 | 28.56 ± 1.11 | 45.65 | 0.70 | 0.29 | 5.54** | 0.59** | 7.53** |
| | 2 | 15.32 | 53.39 | 31.95 ± 0.73 | 26.77 | 0.31 | −0.62 | | | |
| Grain Zn conc. (mg kg$^{-1}$) | 1 | 16.01 | 65.95 | 40.65 ± 0.82 | 23.61 | 0.21 | −0.18 | 3.41** | 0.37** | 4.63** |
| | 2 | 14.75 | 44.54 | 28.91 ± 0.51 | 20.57 | 0.32 | −0.33 | | | |

Notes:
* $p < 0.05$;
** $p < 0.01$.

## Genetic polymorphism of SSR markers

In the study, 38 polymorphic alleles were defined in 138 genotypes using 10 wheat-specific SSR markers. The allele numbers (Na) ranged from 1 (Xgwm154, Xgwm473, and Xwmc177) to 12 (Xgwm282) for each locus (Table 5). The number of effective alleles (Ne) per locus changed from 1.00 to 7.65. Observed (Ho) and expected heterozygosity (He) per locus ranged from 0 to 1.0 (Xgwm515) with an average value of 0. 27, and 0 to 0.87 (Xgwm282) with an average value of 0.44, respectively. The mean I value was 0.88 with the highest amount for the Xgwm282 marker at 2.17 and the lowest for the Xgwm154,

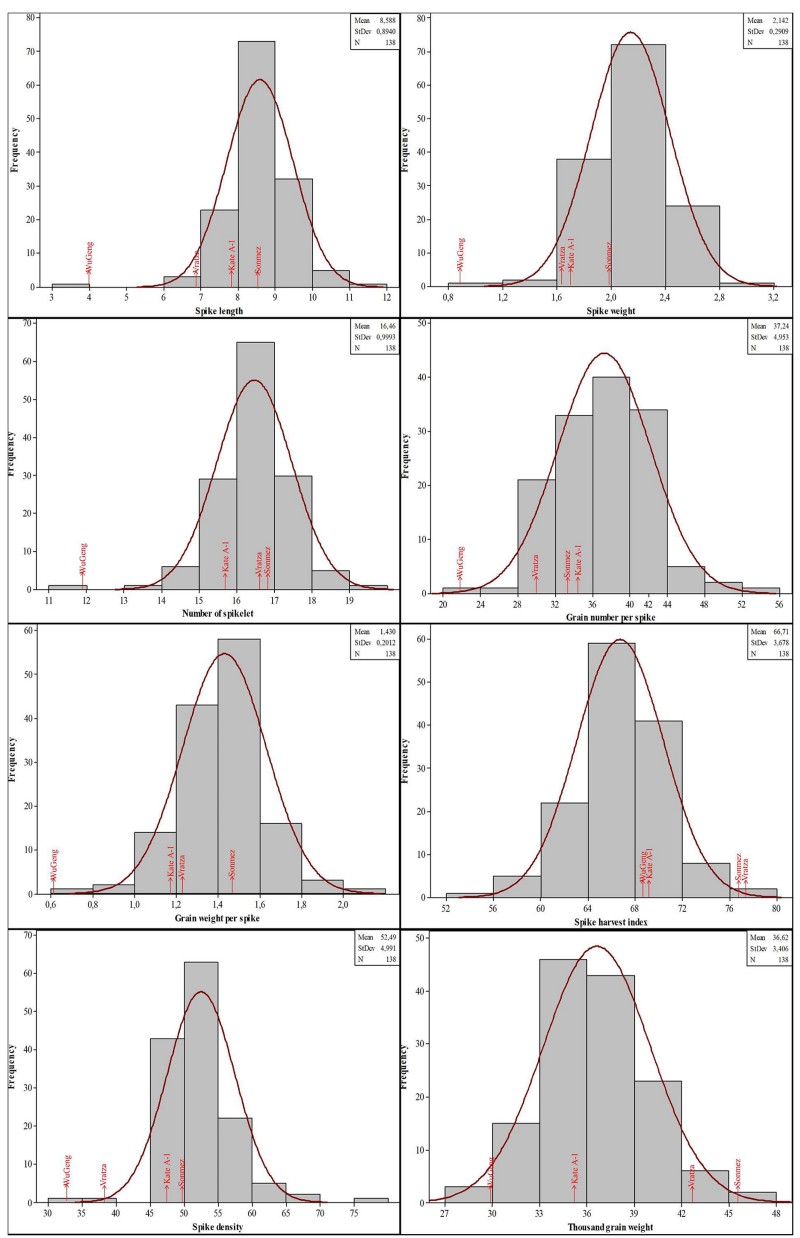

**Figure 2 Distribution of the spike characteristics in the four-parental DH population.**

Xgwm473 and Xwmc177 markers at 0.0. The average PIC value for all markers was 0.45, ranging from 0.0 (Xgwm154, Xgwm473 and Xwmc177) to 0.87 (Xgwm282).

## Genetic diversity and population structure

The 134 DH lines from four-parental $F_2$ hybrids and parental lines were divided into five clusters (Fig. 5). Similarly, the STRUCTURE analysis was also divided into five sub-populations, which was consistent with the UPGMA cluster analysis (Fig. 6). The number of estimated sub-populations (K) was set from 1 to 10 for computing ΔK values, which reached the highest value at K = 5, verifying that the population should be divided into five
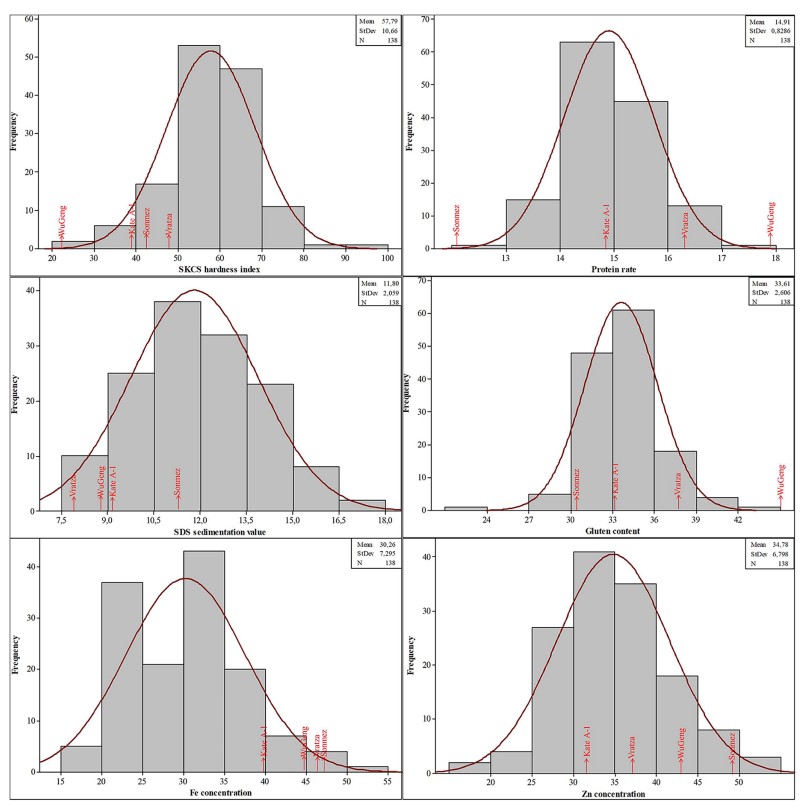

**Figure 3 Distribution of the quality parameters in the four-parental DH population.**

**Table 3 Eigenvalue and contribution of the principal component axes towards variation in the population.**

| Component | Initial | | | Variamax rotation | | |
|---|---|---|---|---|---|---|
| | Eigenvalues | Variance (%) | Cumulative (%) | Eigenvalues | Variance (%) | Cumulative % |
| 1 | 4.35 | 31.08 | 31.08 | 3.58 | 25.56 | 25.56 |
| 2 | 2.08 | 14.83 | 45.91 | 2.18 | 15.56 | 41.12 |
| 3 | 1.75 | 12.53 | 58.43 | 2.12 | 15.13 | 56.25 |
| 4 | 1.38 | 9.83 | 68.26 | 1.39 | 9.95 | 66.20 |
| 5 | 1.02 | 7.30 | 75.56 | 1.31 | 9.36 | 75.56 |
| 6 | 0.88 | 6.31 | 81.87 | | | |
| 7 | 0.78 | 5.59 | 87.46 | | | |
| 8 | 0.58 | 4.15 | 91.61 | | | |
| 9 | 0.48 | 3.45 | 95.07 | | | |
| 10 | 0.40 | 2.84 | 97.91 | | | |
| 11 | 0.18 | 1.26 | 99.17 | | | |
| 12 | 0.11 | 0.78 | 99.95 | | | |
| 13 | 0.01 | 0.03 | 99.99 | | | |
| 14 | 0.002 | 0.01 | 100.00 | | | |

**Table 4 Contribution of studied traits of the mapping population towards major principal components.**

| | Component | | | | |
|---|---|---|---|---|---|
| | 1 | 2 | 3 | 4 | 5 |
| Spike length (cm) | 0.491 | −0.165 | 0.797 | 0.080 | 0.038 |
| Spike weight (g) | 0.886 | −0.083 | 0.279 | 0.042 | 0.111 |
| Number of spike | 0.841 | −0.038 | 0.052 | −0.223 | −0.113 |
| Grain number per spike | 0.890 | −0.231 | −0.133 | −0.061 | 0.164 |
| Grain weight per spike (g) | 0.927 | −0.166 | −0.025 | 0.127 | 0.152 |
| Spike harvest index (%) | 0.190 | −0.217 | −0.782 | 0.244 | 0.091 |
| Spike density | 0.047 | −0.196 | 0.867 | 0.222 | 0.168 |
| Thousand grain weight (g) | −0.047 | 0.206 | 0.029 | 0.746 | −0.150 |
| SKCS value (HI) | 0.172 | −0.016 | 0.077 | 0.018 | 0.743 |
| Protein rate (%) | −0.181 | 0.901 | −0.043 | 0.020 | −0.087 |
| SDS sedimentation value (ml) | −0.076 | 0.331 | 0.017 | −0.488 | 0.225 |
| Gluten content (%) | −0.107 | 0.867 | −0.081 | 0.104 | −0.140 |
| Grain Fe concentration | −0.004 | 0.150 | 0.010 | 0.354 | −0.711 |
| Grain Zn concentration | −0.271 | 0.487 | 0.075 | 0.525 | 0.199 |

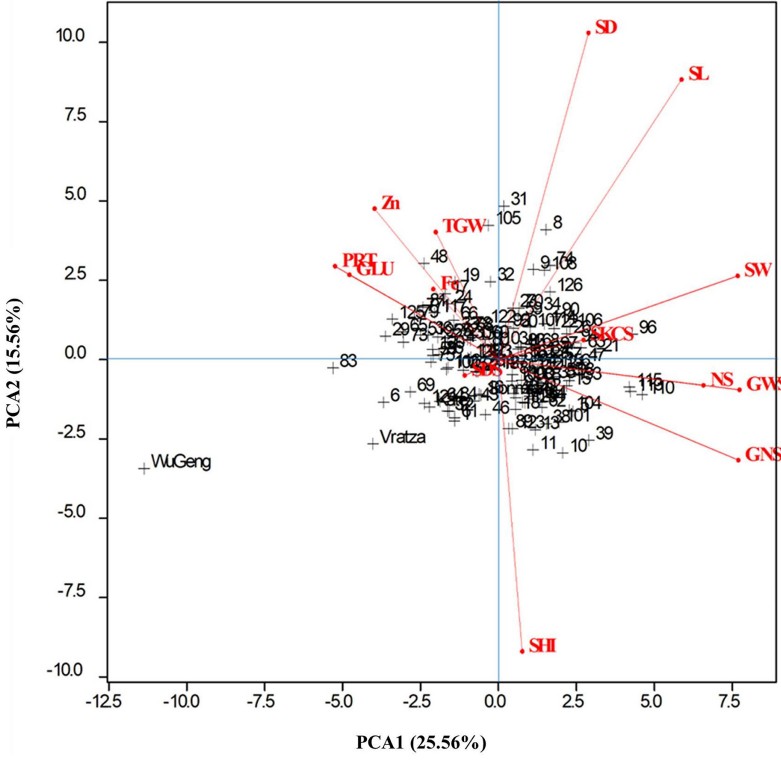

**Figure 4 Genotype-by-trait biplot graph showing 138 genotypes in two main principal components for studied traits.**

**Table 5  The genetic parameters of SSR markers.**

| SSR markers | Na | Ne | I | Ho | He | PIC |
|---|---|---|---|---|---|---|
| Gwm445 | 3 | 2.93 | 1.09 | 0.00 | 0.66 | 0.66 |
| Xbarc108 | 4 | 3.41 | 1.29 | 0.00 | 0.71 | 0.71 |
| Xbarc146 | 6 | 3.31 | 1.48 | 0.97 | 0.70 | 0.70 |
| Xbarc149 | 2 | 1.22 | 0.33 | 0.04 | 0.18 | 0.23 |
| Xbarc61 | 6 | 5.71 | 1.76 | 0.69 | 0.82 | 0.82 |
| Xgwm154 | 1 | 1.00 | 0.00 | 0.00 | 0.00 | 0.00 |
| Xgwm282 | 12 | 7.65 | 2.17 | 0.00 | 0.87 | 0.87 |
| Xgwm473 | 1 | 1.00 | 0.00 | 0.00 | 0.00 | 0.00 |
| Xgwm515 | 2 | 2.00 | 0.69 | 1.00 | 0.50 | 0.50 |
| Xwmc177 | 1 | 1.00 | 0.00 | 0.00 | 0.00 | 0.00 |
| Mean | 3.80 | 2.92 | 0.88 | 0.27 | 0.44 | 0.45 |

sub-populations. The number of genotypes assigned to each sub-population was 25, 33, 45, 10, and 25. The 16 genotypes in the mixed group also included the parent Vratza. Relative kinship coefficients between individuals gained from the Q-matrix showed that about 28.14% of the pairwise kinship coefficients ranged from 0.00 to 0.05.

## Marker-trait association analysis

Results of simple regression analysis showed that Xbarc61 had a significant relationship with all spike characteristics, and grain Fe and Zn concentration (Table 6). In addition, the Gwm445 correlated with the number of spikelets, grain number per spike, and grain Fe concentration, while the Xgwm282 was associated with spike harvest index, SDS sedimentation value, and grain Fe concentration. The Xgwm473 is related to thousand kernel weight, moreover, Xbarc146 had a significant correlation with spike harvest index, grain hardness, protein rate, gluten content, grain Fe and Zn concentration.

## DISCUSSION

### Phenotypic assessment

Identification of novel allelic diversity is important for a successful breeding or mapping study, and the population to be examined must be tested for phenotypic variation as well as genetic diversity. The significant differences between the genotypes and years in the analysis of variance showed that, the phenotypic variation was caused by both genetic and environmental factors. The phenotypic performance of some genotypes is high in the first year, while others are higher in the second year (Table S1). This performance of genotypes, which varies according to years, caused the genotype × year interaction. With the descriptive statistics calculated in the study, the normal distribution of the values obtained in the population and the high variability according to the CV value emphasize that the studied genotypes are a valuable breeding material, as selection is effective when the magnitude of variability in the breeding population is high (Allard, 1960). This phenotypic variation can be used in different wheat breeding programs to develop high-yielding and

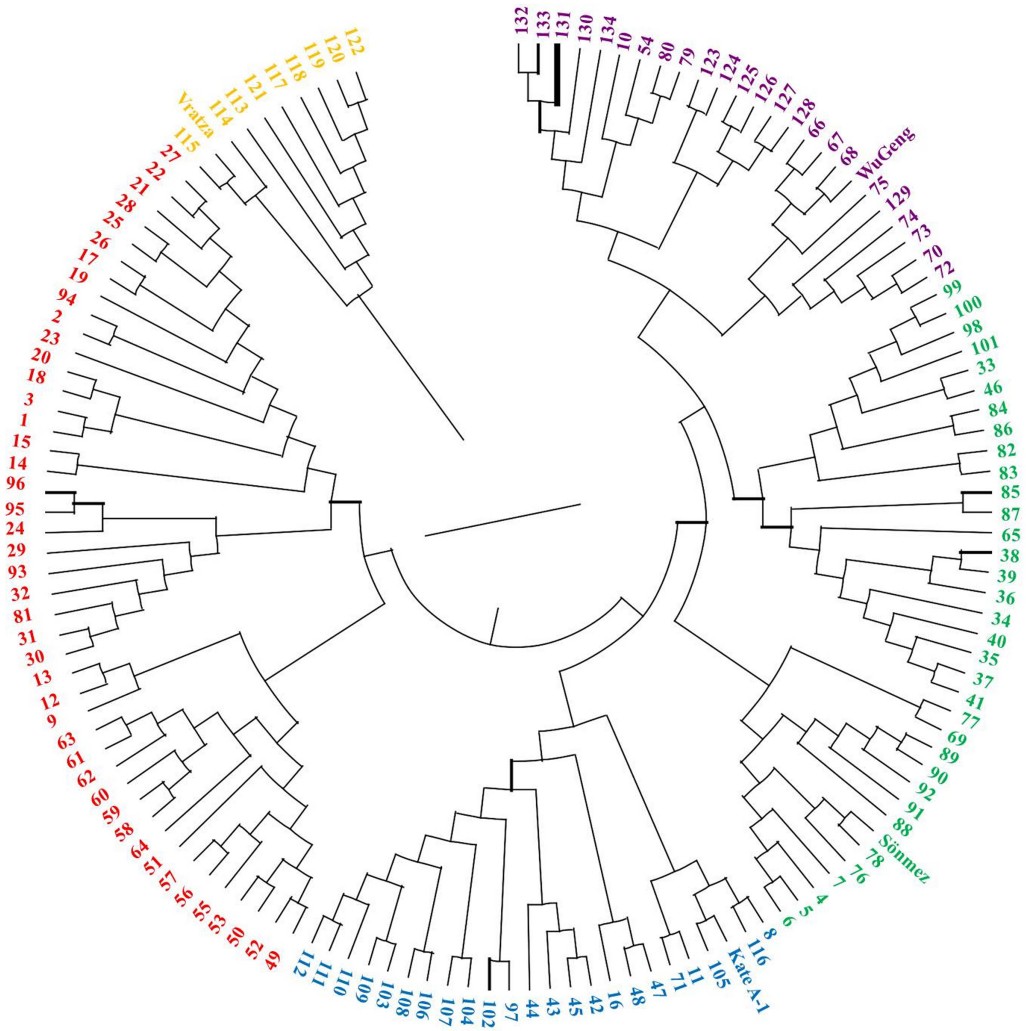

**Figure 5 Dendrogram showing the genetic diversity among 134 DH bread wheat genotypes and four parents based on SSR data.**

quality cultivars. In addition, the variations of phenotypic traits of economic importance play a crucial role in the detection and mapping of desired genes.

The spike traits examined in the study are important phenotypic characters that can be used to improve the yield of common wheat. The findings of this study agree with those of *Kutlu et al. (2017)* and *Kutlu & Sirel (2019)* emphasized the adequate existence of genotypes that can be selected to achieve the desired yield. According to the SKCS grain hardness index, hard wheat scores at least 75, and soft wheat scores at most 30. In this case, six genotypes belong to the hard wheat class in the population. If there is more than 11% protein in the wheat grain, it is classified as good bread-quality wheat (*Karaduman et al., 2015*). All of the genotypes in the study were in the high-quality bread wheat class. The SDS-sedimentation volume of wheat, defined primarily by protein quality, is a rough measure of gluten strength. If its value is above 13.0 ml, it means a strong gluten structure (*Peña et al., 1990*). However, in the Turkish Food Codex Wheat Flour Communiqué, it is

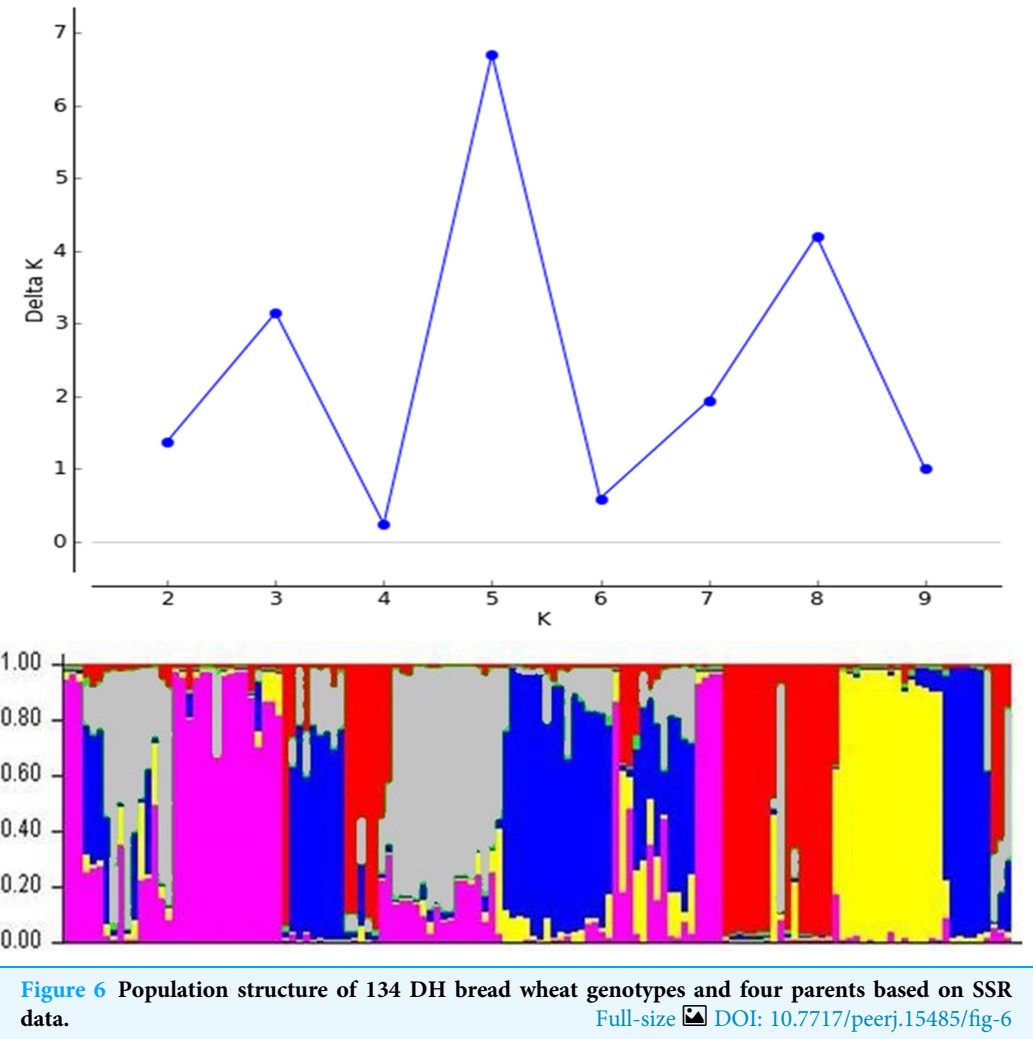

**Figure 6 Population structure of 134 DH bread wheat genotypes and four parents based on SSR data.**

stated that the bread quality of wheat grains with an average of 30% wet gluten is high (*Bulut, 2012*). There are 133 genotypes included in the high-quality bread wheat class in the studied population. Although widely grown modern wheat varieties have a high yield capacity, they are insufficient to supply the daily needs of people in terms of important micronutrient elements such as Fe and Zn. Researchers have reported that wheat grain should contain at least 40 mg/kg Fe and Zn (*Harvestplus Brief, 2006*). In this research, there are 14 genotypes for Fe and 32 genotypes for Zn above this value.

The PCA, a method that identifies the plant traits that contribute most to the variation present within a group of genotypes, assists breeders in the selection of complex genetic characters with low heritability. Spike characteristics, protein and gluten ratios, and Zn concentration are properties that can be used for selection of genotypes. When the bi-plot graph is examined, it can be said that genotypes 19, 29, 48, and 83 can be selected for high quality and genotypes 8, 31, 96, and 105 can be selected for high yield.

**Table 6 Association of marker alleles with studied characteristics in DH bread wheat population detected in simple regression.**

| Trait | B value | The significant marker |
|-------|---------|------------------------|
| SL | 0.006 ± 0.002 | Xbarc61** |
| SW | 0.003 ± 0.001 | Xbarc61** |
| NS | 0.005 ± 0.003 | Xbarc61* |
|  | 0.025 ± 0.008 | Gwm445** |
| GNS | 0.028 ± 0.013 | Xbarc61* |
|  | 0.081 ± 0.041 | Gwm445* |
| GWS | 0.001 ± 0.001 | Xbarc61** |
| SHI | −0.032 ± 0.007 | Xbarc61** |
|  | −0.019 ± 0.007 | Xgwm282** |
|  | 0.055 ± 0.022 | Xbarc146** |
| SD | 0.029 ± 0.013 | Xbarc61* |
| TGW | 0.033 ± 0.018 | Xgwm473* |
| SKCS | −0.126 ± 0.066 | Xbarc146* |
| PR | 0.013 ± 0.005 | Xbarc146** |
| SDS | 0.012 ± 0.004 | Xgwm282** |
| GR | 0.044 ± 0.016 | Xbarc146** |
| Fe | 0.028 ± 0.014 | Xbarc146* |
|  | 0.113 ± 0.020 | Xbarc61** |
|  | −0.030 ± 0.013 | Xgwm282** |
|  | −0.173 ± 0.051 | Gwm445** |
| Zn | 0.105 ± 0.042 | Xbarc146** |
|  | −0.052 ± 0.022 | Xbarc61* |

**Notes:**
* $p < 0.05$;
** $p < 0.01$.

## Genetic polymorphism of SSR markers

The criterion of success in hybridization, which is the first step of breeding studies, is increased genetic diversity. The molecular characterization of a range of wheat genotypes using trait-specific/genic SSR markers and their allelic diversity is important in selecting candidate high-yielding and quality genotypes, selecting parents to develop relevant mapping populations, and initiating wheat breeding programs for this purpose. Trait-specific/genic markers are more effective in detecting true genetic diversity (*Shafi et al., 2022*). The identifiable allele number per locus affects the diversity in a population (*Belete et al., 2021*). In this study, 38 polymorphic alleles ranging from 1 to 12 were detected for each locus, with 7.65 effective alleles. In particular, the Xbarc61 and Xgwm282 were more informative than the others, and they were effective in determining genetic divergence of studied genotypes.

The considerable diversity in the studied population was highlighted by the high mean values of PIC, expected heterozygosity (He), and Shannon information index (I). The high mean I value (0.88) emphasized the genetic richness of the DH population and the wide

genetic base with the hybridizations, selections, and haploidization. The value of Ho in these measures indicated no heterozygosity for Gwm445, Xbarc108, Xgwm154, Xgwm282, Xgwm473, and Xwmc177 loci. Whereas, the He of the Gwm445, Xbarc108, and Xgwm282 loci was quite high. Such a situation may have resulted from the fully homozygous nature of the population. As is common knowledge, observed heterozygosity (Ho), which depends on both the level of genetic variation present in the population and the factors that increase homozygosity, is directly measured from individual genotypes, as opposed to predicted heterozygosity (He), which is estimated from allele frequencies (*Ritland, 1996*). Therefore, by comparing Ho with He, factors that increase homozygosity can be determined (*Schmidt et al., 2021*). It is anticipated that He will be substantially higher in the presence of such factors. The mean PIC value of 0.45 displayed that the markers were informative and efficient to describe the allelic diversity of the population. The markers Xgwm154, Xgwm473, and Xwmc177 did not have allelic diversity, and Xbarc149 had low allelic diversity. The results of genetic parameters revealed that the DH wheat genotypes could be distinguished by six SSR markers that they are Gwm445, Xbarc108, Xbarc146, Xbarc61, Xgwm282, and Xgwm515 showing a high level of polymorphism. Similar results have been reported by other researchers using these SSR markers (*Wang et al., 2012*; *Rafeipour et al., 2016*; *Ahmed et al., 2020*; *Shamsabadi et al., 2022*).

## Genetic diversity and population structure

Due to its homozygosity, DH wheat can be used as a variety, a parent in the creation of new hybrids, and source of genetic mapping studies. Therefore, determining the genetic diversity of DH wheat lines based on phenotypic traits and DNA polymorphisms is significant to determine strategy in breeding programs (*Barakat et al., 2013*). It is recommended that the parents are chosen for hybridization from clusters that are far apart in order to obtain greater genetic diversity in the segregating generations and more heterosis in $F_1$ generation. It is understood from the dendrogram obtained based on the genetic distance that each of the parents used in this study was located in different clusters and were genetically distant from each other. In the first cluster, there were nine DH lines and the parent Vratza, which was the first parent to start hybridization. The similarity of a small number of lines with Vratza may be due to the decrease in its share in the genetic structure. Another parental line, Sönmez, which was lastly included in the hybridization, was in the same cluster with the most DH lines compared to the others. Sönmez's contribution to the genetic structure is expected to be the highest in this hybridization scheme. However, the contribution of new gene combinations that may occur in the $F_2$ segregating generation to genetic diversity should not be forgotten. These segregations may create different variations than expected. Both UPGMA and STRUCTURE analysis divided the genotypes into five. Each parent was in a different group as expected. Some of the DH lines were included in each parent's group, while 44 DH lines were included in a different group. This may prove that traits that are quite different from the parents can emerge in the lines. The DH population studied is a four-parental population with a long history of hybridization and selection. Doubled haploid populations obtained by this type of hybridization have higher variation than bi-parental DH populations because they

contain the characteristics of all four parents in certain proportions. The diversity of the population originated from both successive crosses and the variation that may occur in the $F_2$ segregation generation.

## Marker-trait association analysis

Identifying the relationships between phenotypic traits and molecular markers is a basic step for mapping studies, genomic and marker-assisted selection. In some studies, the association of SSR markers with phenotypic traits has been determined by simple or stepwise regression analysis when the traits to which the markers are associated or linked are known (*Moradi et al., 2014*; *Babay et al., 2019*; *Farhangian-Kashani et al., 2021*). Simple regression analysis was conducted to determine marker-trait associations for spike characteristics, quality traits and bio-fortification capacity of DH wheats with trait-specific/genic SSRs. The aim was to validate already known to be associated with mentioned traits and/or identify new marker-trait association, if any. The results of statistically significant marker-trait associations are presented in Table 6.

Spike characteristics and thousand kernel weight were associated with locus Gwm445, Xbarc61, Xgwm282, and Xgwm473. These markers were associated with grain weight, spikelet number per spike, thousand kernel weight, and grain yield in previous studies (*Kuchel et al., 2007*; *Mir et al., 2012*; *Guo et al., 2015*; *An et al., 2022*; *Kumar, Gill & Nagarajan, 2022*).

The SKCS grain hardness index, protein and gluten rates are associated with Xbarc146 loci, while the SDS sedimentation value is associated with Xgwm282. *Muqaddasi et al. (2020)* reported QTLs for grain hardness and protein rate on chromosomes 2B, 3B, 5D, and 6A. It indicates the effect of environment on these traits, therefore; different QTLs were identified in another environment. In previous studies, these markers have also been associated with spike fertility index, kernel length, grain Fe and Zn concentration (*Krishnappa et al., 2018*; *Chen et al., 2019*; *Li et al., 2022*). Grain Fe concentration is associated with four SSR markers namely Xbarc146, Xbarc61, Xgwm282, and Gwm445, while grain Zn concentration associated with Xbarc146 and Xbarc61. *Zhou et al. (2020)* reported 29 unique loci located on chromosomes 1B, 3B, 3D, 4A, 5A, 5B, and 7A associated with grain Zn concentration in a study of 207 bread wheat varieties. Other aforementioned markers were also reported by *Li et al. (2022)* to be mostly associated with the nutritional quality of wheat.

By the analyzing of a DH population, previous marker-trait associations were verified and some new associations of markers were discovered. Four SSR markers were found to be associated with yield-related spike characteristics, quality traits, and grain Fe and Zn concentrations, which are important for bio-fortification. Moreover, each marker was associated with more than one trait. These results provide clues for the joint heredity of these QTLs and pointed out that the QTLs managing these traits may be co-localized and thereupon more opportunity for coinheritance. Thereupon, increasing not only the concentration of grain Fe and Zn but also yield and quality parameters can easily be improved simultaneously. It can be said that these markers (Xbarc61, Xbarc146, Gwm445,

and Xgwm282) associated with the studied phenotypic features can be used effectively for MAS breeding programs to aim to improve and modify these features.

## CONCLUSIONS

The DH genotypes examined in this study showed a wide range of genetic diversity as well as phenotypic variation. It has been shown that DH populations from four-parent $F_2$ crosses with a long history of hybridization and selection may have a very different genetic structure than expected. Efforts to increase bio-fortification and nutritional quality in staple foods such as wheat reveal the necessity of understanding the genetic mechanism that governs these characteristics. Therefore, molecular screening of breeding lines can be done to better understand the parameters that reveal both the yield and nutritional quality of wheat. The acquisition of marker data from this study can be functional in selecting proper primers and identifying marker association with phenotypic features allowing determining desirable features for high grain yield and quality genes and using them as markers. However, the time-consuming and costly provision of populations suitable for mapping (either segregated or DH) and the lack of proper linkage between phenotypic features and molecular markers are two of the major limitations. A simple association analysis to be used in a validation study with a small number of trait-specific molecular markers makes it possible to overcome these limitations and, using molecular markers together with simple regression analysis, the genetic diversity of DH populations can be used to identify trait-associated gene positions. The results of this study may provide researchers with a new perspective on the evaluation of DH populations and the validation of molecular markers.

## ACKNOWLEDGEMENTS

The authors thank MST Biotechnology Laboratory and Specialist Molecular Biologist Enver Secer for their technical support in conducting molecular analyzes.

### Funding

This research is supported by the Scientific Research Projects Commission of Eskisehir Osmangazi University (project no: 201823D19). The funders had no role in study design, data collection and analysis, decision to publish, or preparation of the manuscript.

### Grant Disclosures

The following grant information was disclosed by the authors:
Scientific Research Projects Commission of Eskisehir Osmangazi University: 201823D19.

### Competing Interests

Imren Kutlu is an Academic Editor for PeerJ.

## Author Contributions

- Imren Kutlu conceived and designed the experiments, performed the experiments, analyzed the data, prepared figures and/or tables, authored or reviewed drafts of the article, and approved the final draft.
- Sadettin Çelik performed the experiments, analyzed the data, prepared figures and/or tables, and approved the final draft.
- Yaşar Karaduman performed the experiments, authored or reviewed drafts of the article, and approved the final draft.
- Özcan Yorgancılar performed the experiments, authored or reviewed drafts of the article, and approved the final draft.

## Data Availability

The raw data is available in the Supplemental File.

## Supplemental Information

Supplemental information for this article can be found online at http://dx.doi.org/10.7717/peerj.15485#supplemental-information.

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
