# Peer review of "Phenotypic and genetic diversity of doubled haploid bread wheat population and molecular validation for spike characteristics, end-use quality, and biofortification capacity"

_PeerJ, doi:10.7717/peerj.15485_

## Round 0.1 · original submission · Minor Revisions

Dear Authors

The manuscript needs a minor revision to be reconsidered for publication. The authors are invited to revise the paper considering all the suggestions made by the reviewers. Please note that requested changes are required for publication.

With Thanks

·

Basic reporting

1. The manuscript needs moderate English grammatical and structural polishing.
2. Authors must add more information about the crop, including its economic importance.
3. Kindly shorten the length of the sentence... lines 49, 51. Also, note that a sentence should not occupy more than two lines. Kindly check throughout the manuscript.

Experimental design

1. Mention the software name used for plotting histogram plots.

Validity of the findings

1. Authors need to crosscheck the L193 with table 2.
2. L229, crosscheck the average value of Xgwm515 with table 5.

Additional comments

1. Kindly mention the abbreviations used throughout the manuscript.
2. L171 and L496, please check the cited reference year in the text and end of the manuscript.

Reviewer 2 ·

Basic reporting

The article complies with the criteria expressed by your journal in terms of Basic reporting.

Experimental design

The article complies with the criteria expressed by your journal in terms of Experimental design.

Validity of the findings

The article complies with the criteria expressed by your journal in terms of Validity of the findings.

Additional comments

I suggest that the years of the study be written in the material and method section (Line 108).
Zadoks growth scale value can be given for the periods specified in line 119.
The number of spikes is written in line 215, it should be corrected as number of spikelet.
I suggest literature support for the explanations in the section starting with "with the descriptive..." between lines 261-264.
It would be appropriate to review the publications that do not have a DOI number in the Reference Section .

The study named "Phenotypic and genetic diversity of doubled haploid wheat population and molecular validation for spike characteristics, end-use quality, and biofortification capacity" is an original study containing a extensive data set. The subject has been taken in many aspects and the results have been sufficiently discussed. In addition, the manuscript is clearly written in professional, unambiguous language. I think that these results will be an important resource for researchers working with plant breeding.

Reviewer 3 ·

Basic reporting

Please find attached manuscript file for specific comments, edits and queries.

Experimental design

Please find attached manuscript file for specific comments, edits and queries.

Validity of the findings

Please find attached manuscript file for specific comments, edits and queries.

Additional comments

Please find attached manuscript file for specific comments, edits and queries.

Annotated reviews are not available for download in order to protect the identity of reviewers who chose to remain anonymous.

---

## Round 0.2 · accepted · Accept

Dear Authors,

I am pleased to inform you that after the last round of revision, the manuscript has been improved a lot, and it can be accepted for publication.

Congratulations on accepting your manuscript, and thank you for your interest in submitting your work to PeerJ.

With Thanks